# Gastroesophageal reflux disease in Sri Lanka: An island-wide epidemiological survey assessing the prevalence and associated factors

**Nilanka Wickramasinghe**[1]*, **Ahthavann Thuraisingham**[2], **Achini Jayalath**[2], **Dakshitha Wickramasinghe**[3], **Dharmabandhu N. Samarasekera**[3], **Etsuro Yazaki**[4], **Niranga Manjuri Devanarayana**[5]

1 Department of Physiology, Faculty of Medicine, University of Colombo, Colombo, Sri Lanka, 2 Ministry of Health, Nutrition and Indigenous Medicine, Colombo, Sri Lanka, 3 Department of Surgery, Faculty of Medicine, University of Colombo, Colombo, Sri Lanka, 4 Gastrointestinal Physiology Unit, Barts and The London School of Medicine, London, United Kingdom, 5 Department of Physiology, Faculty of Medicine, University of Kelaniya, Ragama, Sri Lanka

* nilanka@physiol.cmb.ac.lk

## Abstract

Gastroesophageal reflux disease (GERD) is commonly encountered in clinical practice in Sri Lanka. However, its prevalence in Sri Lanka is unknown. Our objective was to study the island-wide prevalence of GERD symptoms in Sri Lanka and its associated factors. A total of 1200 individuals aged 18–70 years (male: female 1: 1.16, mean age 42.7 years [SD 14.4 years]). were recruited from all 25 districts of the country, using stratified random sampling. An interviewer-administered, country-validated questionnaire was used to assess the GERD symptom prevalence and associated factors. Weight, height, waist, and hip circumference were measured. Heartburn and/or regurgitation at least once a week, an internationally used criterion for probable GERD was used to diagnose GERD. In this study, GERD symptom prevalence was 25.3% (male 42.1% and female 57.9%). Factors independently associated with GERD were inadequate sleep, snacking at midnight, sleeping within two hours of consuming a meal, skipping breakfast, increased mental stress, and certain medications used such as statins, and antihypertensive medications (p<0.001, univariate and logistic regression analysis). 38.4% of the study population have been using medication for heartburn and regurgitation in the past 3 months and 19.8% were on proton pump inhibitors. To conclude, the prevalence of GERD symptoms in Sri Lanka (25.3%) is higher than its estimated global prevalence of 13.8%. Several meal-related lifestyle habits, mental stress, and the use of some medications are significantly associated with GERD, indicating the importance of lifestyle modification and stress reduction in its management.

**Data Availability Statement:** The data has been submitted to Harvard dataverse. Link given below: https://doi.org/10.7910/DVN/D6AOWX.

**Funding:** NW was awarded two grants to for data collection for her PhD research. This does NOT include publication fees. UGC/VC/DRIC/PG2019(1)/CMB/01 University Grants Commission https://www.ugc.ac.lk/ AP/3/2/2020/SG/11 Small Grants, University of Colombo https://cmb.ac.lk/ Funders did not play any role in the study design, data collection and analysis, decision to publish, or preparation of the manuscript.

**Competing interests:** The authors have declared that no competing interests exist.

## Introduction

Gastroesophageal reflux (GER) is defined as the effortless passage of gastric contents into the esophagus. In the majority, it is a normal physiological process. According to the American College of Gastroenterology (ACG) guidelines, when GER leads to troublesome symptoms such as frequent heartburn, regurgitation, vomiting, etc., and when complications such as esophagitis or stricture formation occur, it is referred to as gastroesophageal reflux disease (GERD) [1, 2].

GERD is one of the most common motility disorders affecting all ages and gives rise to a host of unpleasant symptoms such as chest pain, heartburn, acid taste in the mouth, halitosis, vomiting, dysphagia, and respiratory problems. GERD leads to a range of complications such as Barrett's esophagitis, strictures, and extra-esophageal manifestations such as asthma, chronic cough, and laryngitis [3]. The most recent systematic review and meta-analysis on GERD by Nirwan and his team in 2020, analyzing 102 studies, reported a global prevalence of 13.98%, with the lowest prevalence in China (4.16%), and the highest in the USA and UK (21.04%) [4].

GERD is a complex multifactorial disease associated with many genetic and environmental risk factors [5]. The main underlying pathophysiological mechanism leading to GERD is the malfunctioning of the lower esophageal sphincter, commonly triggered by risk factors such as obesity, smoking, genetic predisposition, pregnancy, hiatal hernia, reduced gastric motility, medications, etc. [4]. The diagnosis of GERD usually starts with a clinical history taking. Symptom-based diagnosis can be optimized by using validated, country-specific GERD screening tools with credible sensitivities and specificities. There is a previously validated GERD screening questionnaire available in Sri Lanka. It was developed using a case-control study of 100 GERD patients versus 150 healthy controls, confirmed by endoscopy and/ or pH studies [6].

Sri Lanka, an island spanning 65,610 km$^2$, is situated just south of the Indian subcontinent and has a population of approximately 22 million. Although GERD is believed to be common in Sri Lanka, there are no epidemiological studies assessing its prevalence at the district or national level. A study assessing the association between GERD and asthma reported a GERD prevalence of 28.5% in 202 non-asthmatic controls and 60% in 202 subjects with asthma [7], and a previous hospital-based study conducted by Navarathne et al., has estimated it to be around 15% [8], indicating the probability of a high disease burden in Sri Lanka. In addition, there is no comprehensive evaluation of factors associated with GERD in Sri Lanka.

Treating GERD, which is considered one of the most common non-communicable diseases, is a costly endeavor that also drains the quality of life of these sufferers leading to poor productivity.

It is very important to know the exact prevalence of GERD in Sri Lanka according to different regions, to distribute gastroenterology specialists, investigation facilities such as endoscopy, pH impedance, and esophageal manometry and medications used for GERD, and facilities to conduct surgical management options (e.g., fundoplication). Without a thorough knowledge of the burden of GERD in different regions of the country, it is very difficult to do the above and our current study aims to fill this gap.

We aim to find the demographic, lifestyle, and food-related factors that significantly increase GERD-related symptoms such as heartburn and regurgitation. Based on our clinical experience and findings of previous studies, many factors suggested as triggering factors for GERD may not have an actual relationship with true GERD [4]. Symptoms such as heartburn and regurgitation are recognized as presenting complaints of several diseases other than

GERD, such as functional gastrointestinal disorders (e.g., functional heartburn, functional dyspepsia), esophageal motility disorders (achalasia), and gastroparesis.

Therefore, identifying true association factors and avoiding or preventing them, and modifying these risk factors through lifestyle management strategies is an integral part of the management of GERD.

## Methods and materials

### Study setting

A cross-sectional study representing all 25 districts in all 9 provinces of Sri Lanka was carried out. The study population was adult Sri Lankans, with the inclusion criteria being those of age ranging from 18 to 70 years. Bedbound and wheelchair-bound patients and those who have undergone gastrointestinal surgery were excluded.

### Instruments used in data collection

An interviewer-administered questionnaire was used to gather information. This was available in all three main languages of Sri Lanka and the data collectors' team was proficient in all three local languages (i.e., English. Sinhala, and Tamil). This questionnaire consisted of; a section on demographics and factors associated with GERD symptoms, a country-validated GERD screening tool [6], a country-validated food frequency questionnaire (FFQ) [9], a country-validated International Physical Activity Questionnaire (IPAQ) [10], and a culturally-validated Perceived Stress Scale (PSS) [11, 12].

The section on demographic and socio-economic characteristics also questioned factors associated with/or likely to be associated with GERD symptoms such as alcohol consumption, smoking, betel chewing, etc. Chronic medical conditions such as ischemic heart disease, chronic kidney disease, asthma, and diabetes mellitus, as well as the medications used, were based on self-claims by the participants. Regular use of certain medications was questioned. Quick eating (consuming a meal in 10 minutes), skipping breakfast (>3 times per week), consuming snacks in the middle of the night (>3 times per week), and general quality of sleep whether inadequate or not, were some factors questioned. A comprehensive literature review was carried out studying similar research and systematic reviews done worldwide on GERD and control groups. Factors deemed to be associated with GERD symptoms in these studies were identified and used in the questionnaire [4].

The GERD screening tool assessed the frequency and severity of 7 common GERD symptoms on a Likert scale [6]. The symptoms listed are heartburn, regurgitation, chest pain, dysphagia, cough, bloating, and belching which are considered in the symptom profile of GERD. The World Health Organization (WHO) Stepwise Approach to Surveillance (STEPS) protocols for anthropometric measurements were used to measure weight, height, abdominal circumference, and hip circumference [13]. SECA 813 Portable Weighing scale, SECA 213 Portable Stadiometer, and standard measuring tapes were used for this purpose.

### Definition of GERD used

The "cases" are those who have probable GERD, defined as heartburn and/or regurgitation at least once a week, which is an internationally used criterion for probable GERD (given the difficulty in using invasive investigations to diagnose GERD in the community) [4].

The participants, who did not fall under the above definition of probable GERD, are considered as controls.

## Sample size

The total sample size was 1200. It was calculated using a standard statistical formula described by Lachenbruch et al., 1991, with a 0.05 level of precision [14]. The exact value of the prevalence of GERD in the community is not known in Sri Lanka. Therefore, this proportion was considered as 50% in the calculation to obtain the highest sample size. With an allowance for design effect cluster sampling of 2 and an anticipated non-response of 5%, the required minimum sample size was rounded off to 810. Since this study intended to cover every 25 districts of the country, the sample size was increased to 1200 to have 40 clusters distributed amongst the 25 districts, considering the probability proportionate to the size (PPS) of the population.

## Data collection method

The 2012 census data was used as it is the most recently available population data from the Sri Lanka Department of Census and Statistics. The population data for the whole island and each district was used to identify the number of clusters per district, depending on probability proportionate to population. A group of adults in a Grama Niladhari (GN) division (the smallest administrative unit in Sri Lanka) was defined as a cluster while the cluster size was taken as 30. Clusters were identified using simple random sampling of the list of the GN divisions of each district.

Random selection was used to identify each cluster's index house using a GN division map. Every third house starting from the right of the index house was approached to make up the cluster. After applying inclusion and exclusion criteria only one member from each family was randomly selected by drawing lots. After obtaining informed written consent participants were recruited.

Forty-one subjects who were approached refused or were unable to participate in the study (3.3%). Additional participants were recruited from the same cluster to achieve the sample size. Every third house from the point where recruitment stopped was evaluated until age and sex-compatible willing participant/s were recruited.

## Study duration

The study was conducted from May to July in the year 2021.

## Statistical analysis

The data was analyzed using SPSS 28. Statistical tests carried out on the study population deemed it to have a non-parametric distribution. Chi-square tests for nominal data and Mann-Whitney test for ratio data were used. A backward logistic regression test was used to assess the independent association of factors. The Bonferroni test was carried out when comparing subsets of the population to reduce the issues occurring with multiple comparisons.

## Ethical clearance

The ethical clearance was obtained from the Ethics Review Committee of the Faculty of Medicine, University of Colombo (EC-19-091).

# Results

A total of 1200 participants were recruited from all 25 districts of Sri Lanka (male: female 1: 1.16). All participants completed the questionnaires and were included in the analysis. The mean age was 42.7 years (SD 14.4 years).

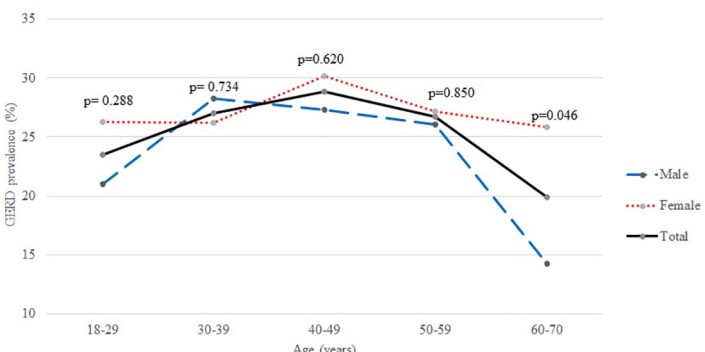

**Fig 1. Prevalence of GERD according to age and sex.** p values for the Chi-square test comparing male and female prevalence per age category.

## Prevalence of GERD in Sri Lanka

In this study, the prevalence of probable GERD (heartburn and or regurgitation weekly) was 25.3% (304). The prevalence of subjects with heartburn weekly and daily is 56.6% and 25.3% while for regurgitation it is 42.2% and 15.8% respectively. Age and sex-specific prevalence of probable GERD is shown in Fig 1.

Fig 2 shows the prevalence of probable GERD across the 25 districts of Sri Lanka. The highest prevalence was noted to be in the districts of Monaragala (40%) and Mullaitivu (40%) while the lowest was in Trincomalee (10%).

## GERD symptoms in total population

Of the total population, 738 (61.5%) had experienced heartburn, 582 (48.5%) had GERD-associated acid/ food regurgitation, and 390 (32.5%) had chest pain, at least once in their lifetime. When questioned about the past month, of the 1200 population, 403 (33.6%) had heartburn, 291 (24.2%) had regurgitation, 191 (15.9%) had chest pain, 314 (26.2%) had bloating, 46 (3.8%) had dysphagia, 89 (7.4%) had a cough and 233 (19.4%) had burping.

## Clinical profile of those with probable GERD

Table 1 demonstrates the GERD-related gastrointestinal symptoms experienced during the past 1 month by the 304 cases. The prevalence of heartburn, regurgitation, chest pain, bloating, dysphagia, cough, and burping was significantly higher in those who fulfilled the criteria for probable GERD, compared to controls (p< 0.001).

## Factors associated with GERD

Following univariate analysis, skipping breakfast, consuming snacks in the middle of the night, inadequate sleep, and high perceived stress scale scores were significantly associated with GERD (Table 2).

A backward logistic regression analysis was performed on the factors listed in Table 2 and daily nutrient intakes (calorie, carbohydrates, protein, fat, fruit, and vegetable portions). Lifestyle factors that were independently associated with GERD were: inadequate sleep [adjusted odds ratio (OR) = 1.586, p = 0.002], snacking at midnight [adjusted OR = 1.720, p = 0.026], sleeping within two hours of consuming a meal [adjusted OR = 1.334, p = 0.038], and skipping breakfast [adjusted OR = 1.753, p <0 .0001]. Certain medications used such as statins

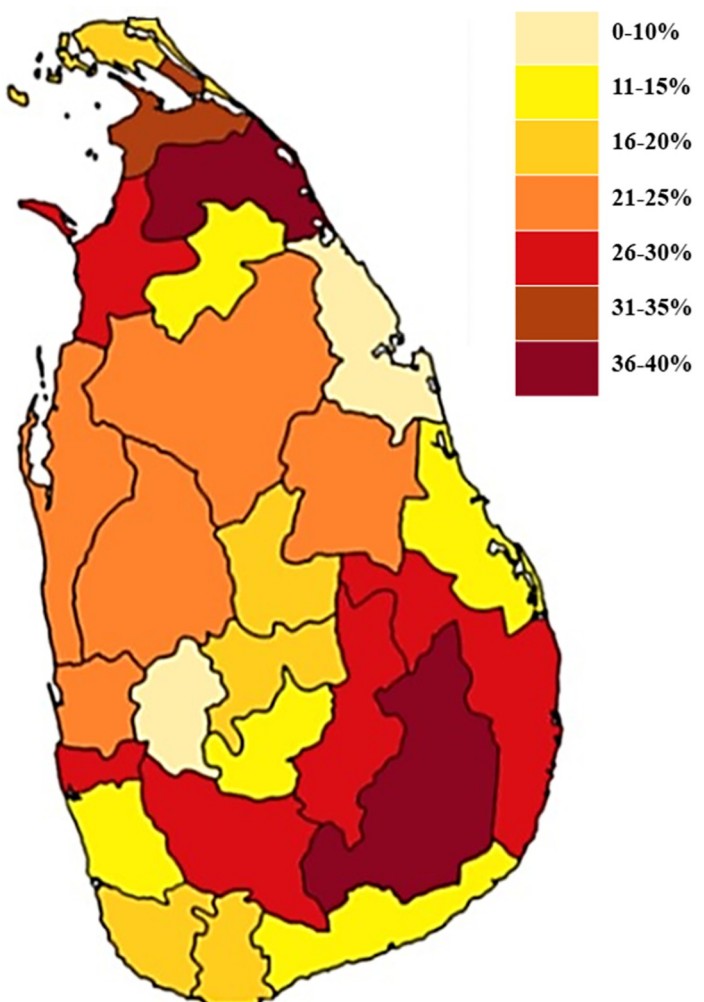

**Fig 2. Prevalence of GERD across the 25 districts of Sri Lanka.** Source of the base layer of the map: Population census layer maps- 2012- Department of Census and Statistics Sri Lanka. www.statistics.gov.lk, available for public domain use.

[adjusted OR = 1.740, p = 0.021] and antihypertensive medications [adjusted OR = 1.508, p = 0.060] were independently associated with GERD. The subtypes of medication were not questioned from the participants. Use of NSAIDs [adjusted OR = 0.576, p = 0.049] and average time seated per day [adjusted OR = 0.954, p = 0.035] were found to be negatively associated with GERD. Here too, increased perceived stress level score was noted to be a significantly associated factor [adjusted OR = 1.957, p < 0.0001].

A separate analysis of the urbanization levels of the cluster samples and the prevalence of probable GERD showed no significant association, meaning that the urbanization of communities did not seem to affect GERD prevalence in Sri Lanka [16].

## Foods that trigger heartburn and regurgitation

Food types that trigger heartburn and regurgitation are listed in Table 3.

**Table 1. Symptom characteristics amongst those with GERD (n = 304).**

| Symptom characteristics | | | n (%) |
|---|---|---|---|
| Presence of GERD symptoms within the last month | | Heartburn | 267 (87.8%) |
| | | Regurgitation | 200 (65.8%) |
| | | Chest pain | 114 (37.5%) |
| | | Bloating | 169 (55.6%) |
| | | Dysphagia | 28 (9.2%) |
| | | Cough | 46 (15.1%) |
| | | Burping | 131 (43.1%) |
| Heartburn | Frequency | Absent | 37 (12.2%) |
| | | Monthly | 18 (5.9%) |
| | | Weekly | 172 (56.6%) |
| | | Daily | 77 (25.3%) |
| | Severity | Mild | 175 (57.6%) |
| | | Moderate | 72 (23.7%) |
| | | Severe | 57 (18.8%) |
| Regurgitation | Frequency | Absent | 104 (34.2%) |
| | | Monthly | 24 (7.9%) |
| | | Weekly | 128 (42.1%) |
| | | Daily | 48 (15.8%) |
| | Severity | Mild | 229 (75.3%) |
| | | Moderate | 54 (17.8%) |
| | | Severe | 21 (6.9%) |
| Chest pain | Frequency | Absent | 190 (62.5%) |
| | | Monthly | 33 (10.9%) |
| | | Weekly | 60 (19.7%) |
| | | Daily | 21 (6.9%) |
| Bloating | Frequency | Absent | 135 (44.4%) |
| | | Monthly | 22 (7.2%) |
| | | Weekly | 103 (33.9%) |
| | | Daily | 44 (14.5%) |
| Dysphagia | Frequency | Absent | 290 (95.4%) |
| | | Monthly | 9 (3%) |
| | | Weekly | 5 (1.6%) |
| | | Daily | 14 (4.6%) |
| Cough | Frequency | Absent | 258 (84.9%) |
| | | Monthly | 7 (2.3%) |
| | | Weekly | 28 (9.2%) |
| | | Daily | 11 (3.6%) |
| Burping | Frequency | Absent | 173 (56.9%) |
| | | Monthly | 15 (4.9%) |
| | | Weekly | 61 (20.1%) |
| | | Daily | 55 (18.1%) |
| Heartburn and / or regurgitation at night-time | | | 158 (52%) |
| Heartburn and / or regurgitation with vomiting | | | 99 (32.6%) |
| Heartburn and / or regurgitation with nausea | | | 109 (35.9%) |
| Heartburn and / or regurgitation with stress | | | 127 (41.8%) |

**Table 2. Comparison of sociodemographic, anthropometric, lifestyle, and health-related parameters between probable GERD and controls.**

| | | Probable GERD | Controls | p-value |
|---|---|---|---|---|
| | | (n = 304) | (n = 896) | |
| **Demographic factors** | | | | |
| Age [mean (SD)] | | 42.42 (13.49) | 42.85 (14.73) | 0.73^ |
| Sex [n (%)] | Male | 128 (42.1%) | 425 (47.4%) | 0.107* |
| | Female | 176 (57.9%) | 471 (52.6%) | |
| Ethnicity [n (%)] | Sinhala | 215 (70.7%) | 612 (68.3%) | 0.26* |
| | Tamil | 68 (22.4%) | 192 (21.4%) | |
| | Muslim | 21 (6.9%) | 90 (10%) | |
| Marital status [n (%)] | Married | 248 (81.6%) | 706 (78.8%) | 0.299* |
| | Single | 56 (18.4%) | 190 (21.2%) | |
| Education [n (%)] | None | 5 (1.6%) | 28 (3.1%) | 0.073* |
| | Up to Grade 5 | 42 (13.8%) | 118 (13.2%) | |
| | Up to Grade 11 | 119 (39.1%) | 349 (39%) | |
| | Up to Grade 13 | 105 (34.5%) | 253 (28.2%) | |
| | University/ Diploma | 29 (9.5%) | 123 (13.7%) | |
| | Postgraduate studies | 4 (1.3%) | 25 (2.8%) | |
| Family income (Sri Lankan Rupees monthly) [n (%)] | <5000 | 11 (3.6%) | 33 (3.7%) | 0.81* |
| | 5000–20000 | 42 (13.8%) | 134 (15%) | |
| | 20000–50000 | 136 (44.7%) | 378 (42.2%) | |
| | 50000–100000 | 87 (28.6%) | 249 (27.8%) | |
| | >100000 | 28 (9.2%) | 102 (11.4%) | |
| **Past medical, surgical, and medication history** | | | | |
| Other chronic disease conditions [n (%)] | Ischemic heart disease | 12 (3.9%) | 34 (3.8%) | 0.905* |
| | Chronic kidney disease | 3 (1%) | 7 (0.8%) | 0.72+ |
| | Diabetes mellitus | 47 (15.5%) | 105 (11.7%) | 0.09* |
| | Asthma | 35 (11.5%) | 71 (7.9%) | 0.057* |
| Medications [n (%)] | Anti-asthmatics | 22 (7.2%) | 45 (5%) | 0.146* |
| | Steroids | 5 (1.6%) | 16 (1.8%) | 0.871* |
| | NSAIDs | 22 (7.2%) | 81 (9%) | 0.332* |
| | Anti-glycemic | 52 (17.1%) | 107 (11.9%) | 0.022* |
| | Antihypertensives | 62 (20.4%) | 134 (15%) | 0.027* |
| | Statins | 50 (16.4%) | 99 (11%) | 0.014* |
| | Other drugs | 45 (14.8%) | 118 (13.2%) | 0.473* |
| Past abdominal surgery [n (%)] | | 78(25.7%) | 200(22.3%) | 0.233* |
| **Lifestyle factors** | | | | |
| Quick eating (consuming a meal in 10 minutes) [n (%)] | | 166 (54.6%) | 492 (54.9%) | 0.926* |
| Skipping breakfast [n (%)] | | 113 (37.2%) | 207 (23.1%) | <0.001* |
| Lying down within 2 hours of having a meal [n (%)] | | 149 (49%) | 378 (42.2%) | 0.038* |
| Sleeping position [n (%)] | Supine | 63 (20.7%) | 133 (14.8%) | 0.056* |
| | Prone | 28 (9.2%) | 72 (8%) | |
| | Right lateral | 56 (18.4%) | 180 (20.1%) | |
| | Left lateral | 52 (17.1%) | 135 (15.1%) | |
| | Combination of all | 105 (34.5%) | 376 (42%) | |
| Number of pillows kept under the head during sleeping [n (%)] | 0 | 4 (1.3%) | 25 (2.8%) | 0.309^ |
| | 1 | 217 (71.4%) | 647 (72.2%) | |
| | 2 | 79 (26%) | 204 (22.7%) | |
| | 3 | 4 (1.3%) | 20 (2.2%) | |

*(Continued)*

**Table 2.** (Continued)

| | | Probable GERD | Controls | p-value |
|---|---|---|---|---|
| | | (n = 304) | (n = 896) | |
| Consuming snacks in the middle of the night [n (%)] | | 36 (11.8%) | 51 (5.7%) | <0.001* |
| Inadequate sleep [n (%)] | | 131 (43.1%) | 264 (29.5%) | <0.001* |
| Smoking [n (%)] | Current | 38 (12.5%) | 105 (11.7%) | 0.065* |
| | Former | 34 (11.2%) | 87 (9.7%) | |
| | Never | 203 (66.8%) | 654 (73%) | |
| | Second hand | 29 (9.5%) | 50 (5.6%) | |
| Daily/ every other day alcohol consumption [n (%)] | | 17 (5.6%) | 33 (3.7%) | 0.15* |
| Tobacco chewing [n (%)] | | 30 (9.9%) | 91 (10.2%) | 0.885* |
| Betel chewing [n (%)] | | 46 (15.1%) | 137 (15.3%) | 0.947* |
| Perceived stress scale score [mean (SD)] | | 13.75 (6.87) | 10.93 (6.80) | <0.001^ |
| Physical activity (MET minutes/week) [mean (SD)] | | 5499.64 (6324.49) | 5128.88 (6565.61) | 0.077^ |
| Average sitting hours per day [mean (SD)] | | 4.43 (3.06) | 4.75 (3.16) | 0.093^ |
| **Anthropometry** | | | | |
| BMI (kg/m2) [mean (SD)] | | 24.89 (4.39) | 24.57 (4.57) | 0.218^ |
| Obesity status + [n (%)] | Underweight & normal | 97 (31.9%) | 345 (38.5%) | 0.038* |
| | Overweight & obesity | 207 (68.1%) | 551 (61.5%) | |
| Abdominal circumference (cm) [mean (SD)] | | 89.08 (12.45) | 88.33 (11.76) | 0.245^ |
| Waist/hip ratio [mean (SD)] | | 0.93 (0.07) | 0.92 (0.07) | 0.126^ |

* Pearson Chi-square

+ Fisher's exact test

^ Mann-Whitney U test

+ According to cut-off values for Asians [15].

Underweight & Normal < 22.9 kg/m2, Overweight > 23 kg/m2

**Table 3. Food items associated with symptoms of heartburn and regurgitation in probable GERD and controls.**

| Food item | Total (n = 1200) | Cases (n = 304) | Controls (n = 896) | p-value | Adjusted OR | 95% CI for OR | |
|---|---|---|---|---|---|---|---|
| | n (%) | n (%) | n (%) | | | Low | High |
| Spicy foods | 399 (33.3%) | 196 (64.5%) | 203 (22.7%) | 0.080 | 0.709 | 0.483 | 1.042 |
| Oily foods | 393 (32.8%) | 212 (69.7%) | 181 (20.2%) | 0.001* | 0.333 | 0.226 | 0.490 |
| Bread | 393 (32.8%) | 207 (68.1%) | 186 (20.8%) | 0.001* | 0.478 | 0.309 | 0.738 |
| Wheat | 305 (25.4%) | 173 (56.9%) | 132 (14.7%) | 0.002* | 0.508 | 0.330 | 0.784 |
| Tea | 146 (12.2%) | 85 (28%) | 61 (6.8%) | 0.475 | 1.190 | 0.738 | 1.919 |
| Coffee | 65 (5.4%) | 47 (15.5%) | 18 (2%) | 0.028* | 0.470 | 0.239 | 0.923 |
| Chocolate | 30 (2.5%) | 19 (6.3%) | 11 (1.2%) | 0.754 | 1.154 | 0.472 | 2.822 |
| Fizzy drinks | 150 (12.5%) | 88 (28.9%) | 62 (6.9%) | 0.052 | 0.647 | 0.417 | 1.003 |
| Sour/ vinegar | 263 (21.9%) | 154 (50.7%) | 109 (12.2%) | 0.001* | 0.510 | 0.346 | 0.752 |
| Milk | 32 (2.7%) | 19 (6.3%) | 13 (1.5%) | 0.516 | 0.758 | 0.329 | 1.749 |

* p<0.05, Logistic regression

**Table 4. Details regarding the use of medications used for heartburn and/ or regurgitation amongst those who used in the total community and amongst those classified as having probable GERD and controls.**

| | | Probable GERD | Controls | Total |
|---|---|---|---|---|
| | | (n = 213) | (n = 248) | (n = 461) |
| | | n (%) | n (%) | n (%) |
| Method of prescribing medication | Doctor prescribed | 112 (52.6%) | 134 (54%) | 246 (53.4%) |
| | Ayurvedic | 4 (1.9%) | 13 (5.2%) | 17 (3.7%) |
| | Over the counter | 31 (14.6%) | 45 (18.1%) | 76 (16.5%) |
| | Both doctor and self-prescribed | 66 (31%) | 56 (22.6%) | 122 (26.5%) |
| Alleviation of symptoms | Complete relief | 65 (30.5%) | 152 (61.3%) | 217 (47.1%) |
| | Partial relief | 144 (67.6%) | 91 (36.7%) | 235 (51%) |
| | No relief | 4 (1.9%) | 5 (2%) | 9 (2%) |
| Type of medication used | Proton pump inhibitors | 133 (62.4%) | 105 (42.3%) | 238 (51.6%) |
| | Histamine blockers | 17 (8%) | 13 (5.2%) | 30 (6.5%) |
| | Domperidone | 34 (16%) | 21 (8.5%) | 55 (11.9%) |
| | Antacids | 63 (29.6%) | 48 (19.4%) | 111 (24.1%) |

Of the whole population, 277 (23.08%), 116 (9.67%), and 28 (2.33%) had symptoms for both bread and wheat products, symptoms for bread only, and symptoms for wheat products only, respectively.

## Treatment

Of the 1200 participants, 461 (38.4%) had used medication targeting heartburn/ regurgitation at least once in their lifetime, whilst 238 (19.8%) had consumed a proton pump inhibitor (PPI) at least once in the past 3 months. Of the 304 probable GERD subjects, 213 (70.1%) used some form of medication for GERD symptoms while the number was only 248 (27.7%) among the controls. Of the 304 probable GERD subjects, 64 (21.1%) were on more than one medication for GERD symptom control. Alleviation of symptoms among those who took medications for heartburn between cases and controls was statistically significant ($p<0.001$).

Table 4 describes details regarding medications used to alleviate symptoms of heartburn etc. amongst the participants.

## Gender differences

When comparing the symptoms of GERD between males and females, only chest pain showed a significant difference ($p<0.001$, male 11.8%, female 19.5%, Pearson Chi-square test), while the prevalence of probable GERD and the GERD score were not significantly different (using Pearson Chi-square and Mann-Whitney U test respectively).

## Discussion

This study is the first island-wide research done to assess the country-wide prevalence of GERD symptoms in Sri Lanka, the factors associated with GERD, and medication use. We calculated the probable GERD prevalence to be 25.3%. Factors such as inadequate sleep, snacking at midnight, sleeping within two hours of consuming a meal, skipping breakfast, increased mental stress, and use of medications such as statins and antihypertensives, are noted to be independently associated with probable GERD. More than one-third of the study population have been using medication for GERD symptoms, while one-fifth were on proton pump inhibitors.

The exact global prevalence of GERD is unknown, despite it being discussed extensively as a common condition. The main reason for this conundrum is the lack of a proper definition for GERD. The gold standard for diagnosing GERD is by combined pH impedance testing [17]. However, the use of this investigation is limited because of the invasive nature of the test and the paucity of facilities available. Thus, most of the epidemiological studies have used a variety of screening tools to assess the symptoms of GERD. The estimated prevalence of GERD ranges from 5% to 25% worldwide [18]. The most recent systematic review and meta-analysis on GERD done by Nirwan et al, using the definition "heartburn and or regurgitation at least once weekly", reported the global prevalence as 13.98% [4].

In our study, we used the same GERD definition, though we used a country-validated GERD screening tool, due to the ease of comparing the results of our study with studies done worldwide.

The prevalence calculated for Sri Lanka by our study (25.3%) is much higher compared to the estimated global prevalence [4]. It is comparable to the GERD prevalence of Turkey (22.40%), which was calculated in the same systematic review to be the highest pooled prevalence of GERD in a country. The lowest GERD prevalence was reported in China (4.16%), while regions such as North America and Europe had a prevalence of 19.55% and 14.12% respectively [4]. Even though several Bangladeshi and Indian studies were quoted in this review, no pooled prevalence was calculated for any South Asian country, to which region Sri Lanka belongs and shares common genetics and ancestry. When the GERD definition of heartburn and/or regurgitation at least once a week was used, a Bangladeshi study reported a prevalence of 5.25% [19], and two Indian studies had a prevalence of 7.6% [20] and 23.6% [21] respectively.

In a small Sri Lankan study on GERD (diagnosed by pH-metry) in asthmatics, the hospital control group had a prevalence of 28.5% while the asthmatic group had a prevalence of 60% [7]. A previous study done by Navarathne et al, using endoscopy in the National Hospital of Sri Lanka estimated the prevalence of GERD to be around 15% [8]. A study done on 1114 advanced level students attending private tuition classes in the Anuradhapura district, noted a probable GERD prevalence of 52% [22]. Thus, though these previous studies were not done in the community, the high prevalence of GERD in our study is compatible with the results reported in some of them.

Except for heartburn or regurgitation, which are the typical GERD symptoms, other symptoms such as chest pain, dysphagia, cough, burping, and bloating, are considered atypical GERD symptoms [2]. The prevalence of these atypical symptoms was wide and varied amongst studies (which used the same GERD definition as ours) worldwide [23, 24]. For example, we found bloating as the most bothersome complaint among those with GERD (55.6%) after heartburn and regurgitation. This contrasts with a study conducted in Korea which reported bloating in only 15.2% of those with GERD [25]. Around 9.2% of our subjects with GERD complained of dysphagia, and this figure was similar to a study in Turkey in which the dysphagia prevalence in those with GERD was 11.9% [26]. In contrast, another study done in China reported dysphagia in 35.7% [27].

We also compared the daily frequency of symptoms and found that the prevalence of those who suffer from daily heartburn and regurgitation in our population is higher than in most studies conducted previously. When comparing the results of a similar study in Turkey to ours in Sri Lanka, the prevalence of daily symptoms of heartburn (1.2%) and regurgitation (1.2%) in Turkey, was lower than the daily symptoms of heartburn and regurgitation reported in our study (25.3% and 15.8% respectively) [26].

Whether the most frequent typical GERD symptom is heartburn or regurgitation is also contradictory in studies. Some studies found heartburn to be more common than

regurgitation as seen in our results [19], whereas other researchers have noted that regurgitation is more common than heartburn [28].

Regarding severity, a study in Turkey reported that 44% of those with heartburn and 28% of those with regurgitation had severe symptoms in contrast to 18.8% and 6.9%, respectively, of our respondents [29]. However, this finding is a very subjective one as severity is a personal perception that is difficult to gauge.

In our study, the mean age of those classified as probable GERD was 42.4 years (SD 13.5 years). GERD prevalence studies around the world show a wide variation regarding association with age. With increasing age, the elasticity of the pharyngoesophageal membrane reduces progressively leading to hiatal hernias which in turn is said to predispose individuals to develop GERD [30]. However, in our study, as seen in Fig 1, the prevalence of GERD symptoms was highest in those of the 40 to 59 years age group, though the values do not show a significant difference between the age categories. This is compatible with the trend shown in the pooled prevalence of GERD in the systematic review by Nirwan et al. [4]. In contrast, a study done in South India, which is the closest to our country, with close genetic ties, noted an increased GERD prevalence in older subjects [31].

Females had a slightly higher prevalence of probable GERD (27.2%) than their male counterparts (23.1%), though the difference was not statistically significant except in the age category of 60 and 70 years. These values are compatible with systematic reviews by Nirwan et al. [4], and Eusebi et al. [32], where the pooled prevalence of GERD was found to be higher in females than in males. Symptoms related to GERD were significantly associated with the female sex in a previous Sri Lankan study [22].

When the prevalence of probable GERD according to education level was investigated, the highest prevalence was seen in those with a medium level of education, followed by those with the lowest level of education while the lowest prevalence was in those with the highest level of education. In the systematic review findings of Nirwan [4] and Eusebi [32], and a South Indian study [31], those with the lowest level of education level had the highest GERD symptom prevalence, whilst those with the highest level of education had the lowest GERD prevalence.

Studies have shown that ethnicities can play a role in developing GERD even in the same country and could be attributed to the diet and lifestyle practices amongst the different ethnicities [33]. Our study, however, did not find any significant difference to show that ethnicity (i.e., Sinhalese, Tamil, Muslim,) plays a factor in developing GERD.

GERD prevalence was also similar between the different economic strata in our study unlike that shown by the recent systematic reviews where the lower income groups reported to have a significantly higher prevalence of GERD [4, 32]. Interestingly, in our study, the consumption of PPI was significantly higher in the low-income groups than in higher-income groups, which has not been reported previously.

The area of residence, being urban or rural, is also considered to play a significant role in the prevalence of GERD because GERD is reported to be higher in urban populations [4, 31]. However, at present, the demarcation of urban and rural is very difficult to assess in Sri Lanka due to the sheer smallness of the country and the high population density. Hence, for measuring urbanization in Sri Lanka, percentage levels of urbanization were proposed as a solution [16]. In our population, no significant difference was noted in the prevalence of GERD symptoms across the various levels of urbanization of the clusters sampled. This could be due to food and lifestyle differences being somewhat similar throughout the country due to Sri Lanka's unique urbanization pattern and small land area [16]. In fact, the highest prevalence was noted in two districts generally perceived as being rural, as opposed to the more urban Colombo district, which is the commercial capital of Sri Lanka.

Certain disease conditions such as asthma, diabetes mellitus, and ischemic heart disease are said to be associated with GERD [34, 35]. Our study did not find a significant association with these diseases. However, we only questioned whether the participants had these diseases, and did not do any investigations or go through their medical records to confirm the presence and severity of these diseases.

Statins and antihypertensives were noted to be significantly associated with GERD in our study. Statins are hypothesized to reduce GERD by decreasing nitric oxide production which reduces the lower esophageal sphincter tone [36]. However, previous epidemiological research has reported controversial results, showing both positive and negative associations [36, 37].

NSAIDs are said to increase gastric acid secretion and mucosal damage contributing to increased GERD symptoms [38]. In our study, the use of NSAIDs was noted to be negatively associated with GERD symptoms, which is contradictory to previous research [4]. One possible reason for our finding might be the regular practice of prescribing of PPIs and other acid-lowering drugs simultaneously with NSAIDs. This is a trend the authors have seen in Sri Lankan medical practice, however with no research done to give concrete evidence.

Of the known factors associated with GERD worldwide, obesity has the strongest evidence for association [4, 39]. The possible reasons for this could be an increased risk of developing a hiatus hernia due to increased intra-abdominal pressure and an increased frequency of transient lower esophageal sphincter relaxation [40]. However, though the values of the markers of obesity such as BMI, and waist-hip ratio were all higher in those classified as probable GERD in our study, it was not statistically significant from that of controls.

Certain habits noted to be associated with GERD, such as sleeping horizontally after consuming a meal [41], skipping breakfast [42], several pillows used to elevate the head [43], different sleeping positions [44], eating a meal quickly [42], having a midnight snack [42], and inadequate sleep [42], were all more common amongst cases than controls. Similarly in our study, inadequate sleep, snacking at midnight, sleeping within two hours of consuming a meal, and skipping breakfast had a statistically significant independent association with probable GERD. This shows that lifestyle modification is an essential component in the management of GERD symptoms.

The association between physical activity and GERD symptoms is complex. Studies have found a moderately strong negative association between the risk of reflux and frequent physical exercise [45]. While one could argue that those who engage in more physical fitness will have lesser obesity and thus a lesser chance of developing GERD, it is also noted that engaging in physical activity increases the chance of reflux, through various mechanisms such as reduced lower esophageal sphincter pressures and delayed gastric emptying [46]. Our study did not have statistically significant findings in this regard.

Alcohol is an associated factor of GERD and increased use of alcohol is noted among those suffering from GERD symptoms [27]. However, our study did not find an association between alcohol and GERD symptom prevalence. The study by Nirwan et al. too did not note such an association [4]. Conversely, while the same study by Nirwan et al. noted a significant association between cigarette smoking and GERD, we did not find such an association. Betel and tobacco chewing is an inherently Asian practice, which is claimed to be increased in GERD patients [31]. However, an association between these practices and GERD symptoms could not be identified in our population.

Studying the association between dietary factors and GERD is difficult due to the number of variables such as variations in quality, quantity, duration, and cooking methods. Furthermore, those with reflux symptoms can tend to avoid various foods that provoke GERD symptoms. Thus, we questioned our population on the food items that generally trigger GERD symptoms. Food sensitivity plays a major role with around one-third of the whole population,

and more than half of the probable GERD group having sensitivity for spices, oily food, and bread (Table 3). This is considerable since Sri Lankans generally eat their food with many spices including a high quantity of chilies. In this study, sensitivity to oily food, wheat, bread, coffee, and sour/vinegar food was significantly associated with GERD symptoms. We found that bread and wheat caused heartburn or regurgitation in 32.8% and 68.1% of the whole population and 25.4% and 56.9% of the probable GERD group, respectively. Gluten-related disorders have more symptoms related to bloating and diarrhea, while the major complaint in our population was heartburn and bloating which makes it unlikely to be gluten [47]. No studies related to finding the prevalence of gluten-related disorders have been done in Sri Lanka and this should be rectified.

A surprising finding in our study was that around 10% of patients specifically indicated that they get heartburn and regurgitation after consuming bread only, but not after consuming any other wheat product. This isolated association between GERD symptoms and bread needs to be evaluated further. The phenomenon of bread-induced GERD symptoms is quite rare in the literature [48]. Further research is needed on this subject to discover if it is related to gluten or related to local bread production in Sri Lanka.

Stress, anxiety, and other psychological factors are proven to be associated with GERD-related symptoms [49]. In agreement, more than one-third of our participants had moderate to severe stress while a similar percentage reported that stress exacerbated their GERD symptoms. After analyzing the perceived stress scale score, a significant association was noted between stress and GERD symptoms. One could hypothesize that a significant number of those who present with GERD symptoms are those with functional heartburn related to stress in Sri Lanka.

Medical treatment of GERD most often uses PPIs and is the first and most common treatment option [50, 51]. One-third of our study population has been using some sort of medication for heartburn and regurgitation in the past 3 months and one-fifth were on proton pump inhibitors. This is an alarming figure and gives rise to the problems of side effects and the high cost of using these medications [52–54]. This is comparable with a Sri Lankan study on 1000 prescriptions dispensed at State Pharmaceutical Corporation, Anuradhapura district, Sri Lanka where PPIs ranked as the most prescribed drug. Studies done in the USA also have shown that 25% to 70% of prescriptions for PPIs had no appropriate indication [50, 51]. Of the 304 participants classified as having GERD in our study, 70.1% were on medication for GERD and 62.4% of them were on PPIs, but only 30.5% reported complete relief from their symptoms with treatment. The authors have noted that in Sri Lanka, the attitude towards acid-lowering drugs is lax and PPIs and other medications can be bought by laypeople themselves without a proper doctor's prescription. This self-medication hinders patients from seeking medical advice and receiving management after an exact diagnosis.

This study has several strengths. The participants were selected using a stratified random sampling technique throughout the entire country (from all 25 districts) and represented the distributions of age, sex, ethnicities, and religions in Sri Lanka. In addition, we used validated standard tools to collect data on GERD symptoms, diet, physical activity, etc., which increased the validity of our findings. Whilst we acknowledge the strengths, there are a few limitations of the study. First, since this is a cross-sectional study, we cannot identify the exact risk factors that caused GERD symptoms and can only assess associated factors. Secondly, recall and responder bias can affect the interpretation of our results. However, as we asked our participants about their GERD symptoms over a short period, we hoped that the bias would be less. Thirdly, we are only using a questionnaire with no endoscopic or pH impedance studies to confirm the diagnosis of GERD. Finally, GERD symptoms can be caused by multiple factors

and have many associated risk factors, all of which cannot be assessed by our study though we attempted to evaluate as many as possible.

In conclusion, the prevalence of GERD symptoms and the use of medication for it is higher in Sri Lanka when compared to studies done worldwide. This indicates a significant health problem that surely compromises quality of life. When managing these symptoms in patients, emphasis should also be given to lifestyle habits and mental stress.

## Supporting information

**S1 Checklist. STROBE statement—Checklist of items that should be included in reports of observational studies.**
(DOCX)

## Author Contributions

**Conceptualization:** Nilanka Wickramasinghe, Dharmabandhu N. Samarasekera, Etsuro Yazaki, Niranga Manjuri Devanarayana.

**Data curation:** Nilanka Wickramasinghe, Ahthavann Thuraisingham, Achini Jayalath.

**Formal analysis:** Nilanka Wickramasinghe, Dakshitha Wickramasinghe, Niranga Manjuri Devanarayana.

**Funding acquisition:** Nilanka Wickramasinghe, Niranga Manjuri Devanarayana.

**Investigation:** Nilanka Wickramasinghe, Ahthavann Thuraisingham, Achini Jayalath, Niranga Manjuri Devanarayana.

**Methodology:** Nilanka Wickramasinghe, Niranga Manjuri Devanarayana.

**Project administration:** Nilanka Wickramasinghe, Ahthavann Thuraisingham, Achini Jayalath.

**Resources:** Nilanka Wickramasinghe.

**Software:** Dakshitha Wickramasinghe.

**Supervision:** Dakshitha Wickramasinghe, Dharmabandhu N. Samarasekera, Etsuro Yazaki, Niranga Manjuri Devanarayana.

**Writing – original draft:** Nilanka Wickramasinghe, Niranga Manjuri Devanarayana.

**Writing – review & editing:** Nilanka Wickramasinghe, Dakshitha Wickramasinghe, Dharmabandhu N. Samarasekera, Etsuro Yazaki, Niranga Manjuri Devanarayana.

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
