## [Decision Letter · Decision Letter 0]

29 Aug 2023

PGPH-D-23-01135

Gastroesophageal Reflux Disease in Sri Lanka: an island-wide epidemiological survey assessing the prevalence and associated factors

Dear Dr. wickramasinghe,

Thank you for submitting your manuscript to PLOS Global Public Health. After careful consideration, we feel that it has merit but does not fully meet PLOS Global Public Health’s publication criteria as it currently stands. Therefore, we invite you to submit a revised version of the manuscript that addresses the points raised during the review process.

Please note that we have only been able to secure a single reviewer to assess your manuscript. We are issuing a decision on your manuscript at this point to prevent further delays in the evaluation of your manuscript. Please be aware that the editor who handles your revised manuscript might find it necessary to invite additional reviewers to assess this work once the revised manuscript is submitted. However, we will aim to proceed on the basis of this single review if possible. 

The reviewer has raised a number of concerns that need attention. They request revisions to improve the quality of reporting, particularly the reporting of the Methods and presentation of the Results.

They have also raised concerns about the structure of the manuscript. Please note that PLOS Global Public Health does not impose restrictions on the structure of manuscripts but the presentation of the manuscript should be logical and enable readability. More guidance can be found at https://journals.plos.org/globalpublichealth/s/submission-guidelines#loc-parts-of-a-submission

We look forward to receiving your revised manuscript.

Kind regards,

Marianne Clemence

Staff Editor

Journal Requirements:

1. Please provide separate figure files in .tif or .eps format and remove them from the manuscript fle.

2.  In the online submission form, you indicated that "The datasets used and/or analyzed during the current study are available from the corresponding author upon reasonable request.". 

3. Some material included in your submission may be copyrighted. According to PLOS’s copyright policy, authors who use figures or other material (e.g., graphics, clipart, maps) from another author or copyright holder must demonstrate or obtain permission to publish this material under the Creative Commons Attribution 4.0 International (CC BY 4.0) License used by PLOS journals. Please closely review the details of PLOS’s copyright requirements here: PLOS Licenses and Copyright. If you need to request permissions from a copyright holder, you may use PLOS's Copyright Content Permission form.

Potential Copyright Issues:

Figure 2: please (a) provide a direct link to the base layer of the map (i.e., the country or region border shape) and ensure this is also included in the figure legend; and (b) provide a link to the terms of use / license information for the base layer image or shapefile. We cannot publish proprietary or copyrighted maps (e.g. Google Maps, Mapquest) and the terms of use for your map base layer must be compatible with our CC-BY 4.0 license. 

4. We have noticed that you have uploaded Supporting Information files, but you have not included a list of legends. Please add a full list of legends for your Supporting Information files after the references list. 

Additional Editor Comments (if provided):

Reviewers' comments:

Reviewer's Responses to Questions

**Comments to the Author**

1. Does this manuscript meet PLOS Global Public Health’s publication criteria? Is the manuscript technically sound, and do the data support the conclusions? The manuscript must describe methodologically and ethically rigorous research with conclusions that are appropriately drawn based on the data presented.

Reviewer #1: Partly

2. Has the statistical analysis been performed appropriately and rigorously?

Reviewer #1: I don't know

3. Have the authors made all data underlying the findings in their manuscript fully available (please refer to the Data Availability Statement at the start of the manuscript PDF file)?

Reviewer #1: Yes

4. Is the manuscript presented in an intelligible fashion and written in standard English?

Reviewer #1: No

5. Review Comments to the Author

Reviewer #1: Dear Authors:

I have read your manuscript. As you have mentioned the prevalence of GERD is rising all over the world, and not only in the western world. As the first country-wide epidemiologic study is important and will be read and cited is global epidemiological studies and as you are filling the missing data is of importance.

As a general and in detail view of the manuscript there are some points to say:

• The manuscript is very long and there are many repeated items in the text that could be omitted

• The main structure is like a research report rather that a medical journal manuscript. You should not tell all of the findings you had found in the study, which were obvious and adds nothing to global knowledge. I think it is better to focus on your best and the most important findings and the items that are different from other parts of the world or alike the other parts that are important to mention.

• I think you have used the term of “reflux” instead of “(acid) regurgitation” interchangeably in the text. The symptom regurgitation differs from the reflux process and reflux disease!

• In the sample enrollment process, there is nothing about how you came to the number of 1200 people and how you made your clusters from the country. You had lost some 50 persons due to some reasons that is obvious, but I do not think that the method you enrolled new people is methodologically correct.

• The term “bloating” as a symptom of GERD disease seems not proper. You may have considered “belching”. If otherwise please give some reference to this symptom in GERD patients.

• Many items in the tables and main text can be shortened to include only the necessary points and facts to be shown.

• In the discussion section there is a section, we it is said that NSAIDs can induce GERD. They are a well-known etiology for upper GI erosions/ ulcers and also dyspeptic symptoms, but not a main factor in the GERD.

• For the figure 1, I think the box plot type may be a better format concerning the age group chosen in the chart

6. PLOS authors have the option to publish the peer review history of their article (what does this mean?). If published, this will include your full peer review and any attached files.

**Do you want your identity to be public for this peer review?** For information about this choice, including consent withdrawal, please see our Privacy Policy.

Reviewer #1: No

---

## [Decision Letter · Decision Letter 1]

16 Jan 2024

PGPH-D-23-01135R1

Gastroesophageal Reflux Disease in Sri Lanka: an island-wide epidemiological survey assessing the prevalence and associated factors

Dear Dr. Wickramasinghe,

Thank you for submitting your manuscript to PLOS Global Public Health. After careful consideration, we feel that it has merit but does not fully meet PLOS Global Public Health’s publication criteria as it currently stands. Therefore, we invite you to submit a revised version of the manuscript that addresses the points raised during the review process.

Please note that we have only been able to secure a single reviewer to assess your manuscript. We are issuing a decision on your manuscript at this point to prevent further delays in the evaluation of your manuscript. Please be aware that the editor who handles your revised manuscript might find it necessary to invite additional reviewers to assess this work once the revised manuscript is submitted. However, we will aim to proceed on the basis of this single review if possible. 

The reviewer has raised a number of concerns that need attention - please see the comments below.

Could you please revise the manuscript to carefully address the concerns raised?

We look forward to receiving your revised manuscript.

Kind regards,

Steve Zimmerman, PhD

PLOS Staff Editor

Journal Requirements:

Additional Editor Comments (if provided):

Reviewers' comments:

Reviewer's Responses to Questions

**Comments to the Author**

1. If the authors have adequately addressed your comments raised in a previous round of review and you feel that this manuscript is now acceptable for publication, you may indicate that here to bypass the “Comments to the Author” section, enter your conflict of interest statement in the “Confidential to Editor” section, and submit your "Accept" recommendation.

Reviewer #2: (No Response)

2. Does this manuscript meet PLOS Global Public Health’s publication criteria? Is the manuscript technically sound, and do the data support the conclusions? The manuscript must describe methodologically and ethically rigorous research with conclusions that are appropriately drawn based on the data presented.

Reviewer #2: Yes

3. Has the statistical analysis been performed appropriately and rigorously?

Reviewer #2: Yes

4. Have the authors made all data underlying the findings in their manuscript fully available (please refer to the Data Availability Statement at the start of the manuscript PDF file)?

Reviewer #2: Yes

5. Is the manuscript presented in an intelligible fashion and written in standard English?

Reviewer #2: Yes

6. Review Comments to the Author

Reviewer #2: Wickramasinghe and colleagues report on an epidemiological survey assessing the prevalence of GERD and associated factors in Sri Lanka. There is limited data from that part of the world so understanding GERD locally is a compelling reason to perform the investigation. The methods are appropriate for the stated aim of the study. However, there are some issues in the paper in its present form that preclude publication with the present version.

Major issues:

Results line 181: For clarity that sentence should mention probable weekly GERD was 25.3%. Do you have weekly heartburn, regurgitation and heartburn + regurgitation prevalence values? That should be included. Was daily overall prevalence along with breakdown above obtained as well, if so that should be included in section as well.

Results line 230-231: antihypertensive medications were independently associated with GERD. Was a sub analysis of antihypertensive types taken done to determine if this finding was associated with a particular drug class (calcium channel blocker, diuretic, beta blocker, ACE or ARB). Believe that is important to know.

Results table 4: In the probable GERD column/ type of medication row box, the total adds up to more than the number in the Probable GERD group suggesting that some patients use multiple medications. Is that number known and if so, it should be included. Also, the control column/ type of medication box, the total does not add up to the number of controls suggesting that some did not take any medication. That should also be reported.

Discussion lines 315-316: This sentence needs to be changed as the comparison between daily heartburn and regurgitation rates in Turkey are compared to weekly in Sri Lanka. This is not a legitimate comparison. The appropriate one is either week to week or daily to daily.

Some minor issues in the revision that should be corrected:

Abstract lines 44-45: sleeping within two hours of consuming a meal is mentioned twice, the second one should be removed.

Introduction line 65: Barret’s should be Barrett’s.

7. PLOS authors have the option to publish the peer review history of their article (what does this mean?). If published, this will include your full peer review and any attached files.

**Do you want your identity to be public for this peer review?** For information about this choice, including consent withdrawal, please see our Privacy Policy.

Reviewer #2: No

---

## [Editor Report · Decision Letter 2]

4 Mar 2024

PGPH-D-23-01135R2

Gastroesophageal Reflux Disease in Sri Lanka: an island-wide epidemiological survey assessing the prevalence and associated factors

Dear Dr. Wickramasinghe,

Thank you for submitting your manuscript to PLOS Global Public Health. After careful consideration, we feel that it has merit but does not fully meet PLOS Global Public Health’s publication criteria as it currently stands. Therefore, we invite you to submit a revised version of the manuscript that addresses the points raised during the review process.

We look forward to receiving your revised manuscript.

Kind regards,

Kenneth Vega, MD

Guest Editor

Journal Requirements:

2. We have noticed that you have uploaded Supporting Information files, but you have not included a list of legends. Please add a full list of legends for your Supporting Information files after the references list.

Additional Editor Comments (if provided):

Wickramasinghe and colleagues have revised the manuscript and addressed all reviewer comments. The manuscript is ready for publication once reference 2 is corrected. The author list for reference 2 should be Vakil N, van Zanten SV etc not N Vakil N, van Zanten SV
---

## [Editor Report · Decision Letter 3]

8 Apr 2024

Gastroesophageal Reflux Disease in Sri Lanka: an island-wide epidemiological survey assessing the prevalence and associated factors

PGPH-D-23-01135R3

Dear Dr Wickramasinghe,

We are pleased to inform you that your manuscript 'Gastroesophageal Reflux Disease in Sri Lanka: an island-wide epidemiological survey assessing the prevalence and associated factors' has been provisionally accepted for publication in PLOS Global Public Health.

Best regards,

Kenneth Vega, MD

Guest Editor

All requested changes have been made and the manuscript is ready for publication.